# Liquid metal interface enables glassy MOF membranes with defect-mediated CO$_2$ transport

Xiaoheng Jin[1], Xing Wu[1], Derrick Ng[1], Aaron W. Thornton [1], Durga Acharya[1], Huanting Wang [2] & Zongli Xie [1] ✉

Glassy metal–organic frameworks (MOFs) combine structural disorder with thermal processability, yet their use as membranes has been hindered by difficulties in fabricating thin, continuous and defect-free films. Here we show a float glass–inspired strategy in which liquid gallium guides the vitrification of ZIF-62 into freestanding glassy MOF membranes. By matching surface energy between melt and bath, dewetting is suppressed, enabling uniform membranes with tunable thickness. In pure glassy MOF membranes, uncoordinated nitrogen sites generated during melting enhance CO$_2$ diffusion, experimentally validating a sorption-assisted transport mechanism. Post-synthetic methylation of these sites reverses CO$_2$/H$_2$ selectivity and raises activation energy. We further identify a glassy impurity phase of ZIF with zni topology that emerges under specific conditions, diminishing CO$_2$ uptake and membrane performance. These results establish how interfacial control and defect engineering together enable high-performance glassy MOF membranes and provide an experimental foundation for probing structure–transport relationships in disordered porous materials.

Metal–organic framework (MOF) glasses are an emerging class of porous materials that combine the modularity of MOF precursors with the processability of amorphous solids[1,2]. Their ability to undergo melting and reshaping offers new opportunities in catalysis, coatings, and separations[3–7]. However, transforming these glasses into continuous and defect-free membranes remains challenging, especially while preserving pore accessibility and structural homogeneity. Most of the reported membranes are either monolithic glass blocks formed by furnace melting or thick composite films supported on inert substrates. These structures are often millimetre-thick and unsuitable for practical separations due to low permeance and poor structural control.

The key limitation arises from the surface properties of the melt. Glass-forming MOFs such as ZIF-62 exhibit high surface tension (~400–500 mN/m) and viscosity (~$10^5$ Pa·s) at molten state, which together promote strong dewetting[8]. This combination resists spreading much like viscous droplets on non-wetting surfaces since high surface tension implies a strong cohesive force within the melt, while high viscosity limits flow and conformal coverage, both unfavourable for uniform film formation. These challenges are further exacerbated by the chemical nature of MOFs that coordination-driven bonding leads to asymmetric environments at solid interfaces, promoting under-coordinated metal nodes and raising interfacial energy[9]. When interacting with porous or rough substrates like alumina, the situation worsens. The nanoscale roughness increases the effective contact angle (Wenzel regime), amplifying dewetting behaviour[10]. In addition, the limited electronic compensation between molten MOFs and oxide surfaces may destabilize the interface, resulting in droplet formation or film rupture. These factors collectively help explain the observed difficulty in forming thin, continuous, defect-free MOF glass coatings on common substrates such as alumina[11,12].

[1]CSIRO Manufacturing, Clayton, VIC, Australia. [2]Department of Chemical and Biological Engineering, Monash University, Clayton, VIC, Australia. ✉e-mail: zongli.xie@csiro.au

To address this limitation, inspired by the float-glass production, we propose the use of liquid metals, such as gallium and gallium alloys, as molten baths for the shaping of MOF glass. These metals provide an ultraflatness[13], relatively inert but electron-rich surface, with surface tensions (>500 mN/m at 673 K) matching those of molten MOFs[14]. Such liquid metals may provide favourable surface polarizability and smooth electron-neutral interfaces, which could help stabilise the molten MOF during spreading[15]. Therefore, we hypothesize that such an interface could promote the formation of uniform, high-permeance pure MOF glass membranes. This concept draws inspiration from the float glass process, where molten silica is cast onto a molten tin bath to achieve extremely flat, continuous sheets. Key principle of float glass fabrication lies in the interfacial tension matching between the liquid bath and the melt, which enables defect-free shaping without mechanical support[16]. Among various liquid metals, gallium stands out due to its low melting point and relative abundance[13]. The passivating nature of gallium oxide layer allows easy recycling of the liquid metal bath at ambient conditions[17]. From an environmental perspective gallium is non-bioaccumulative, chemically stable, and low toxicity, making it suitable as bath material[18].

Fabricating thin, uniform, and phase-pure membranes is not only a technical advancement but a scientific necessity for elucidating structure–transport relationships in porous glass. Such membranes minimise structural ambiguity, a persistent challenge in glassy material research due to the absence of long-range order. In addition to structural clarity, they enable accurate permeation measurements by isolating the influence of local disorder from bulk or interfacial artefacts. This combination of capabilities is particularly critical, given that prior studies have largely relied on indirect structural analysis or simulations to hypothesise the role of impurities or defects in glassy MOFs.

In this context, we investigate how coordination disorder introduced during vitrification and synthetic impurities from precursor-derived crystalline phases influence the gas transport behaviour of glassy MOFs. Simulations and monolithic crystal spectroscopic data suggest that vitrification leads to the collapse of medium-range order[19] and the emergence of local structural irregularities, such as dangling linkers[6]. These local disruptions are believed to enhance guest–framework interactions for quadrupole in molecules such as $CO_2$[20]. However, the direct impact of such defects on membrane-level performance remains poorly understood and lacks experimental validation. Apart from structural defects, there has been growing interest in understanding how synthesis routes influence the structure and properties of crystalline MOFs. However, the role of alternative topologies in shaping the final glass structure and their impact on membrane performance remains insufficiently explored. For example, the ZIF-4 with zni topology has been frequently observed in both laboratory-synthesised and commercial ZIF products[8,21–23]. This ZIF-zni phase can also undergo amorphization, but its effect on the structure and separation behaviour of the resulting glassy membrane has not been investigated. These nuanced variations complicate the understanding of structure–property relationship in glassy ZIF and make it difficult to draw clear conclusions from standalone membrane performance data[11,12,24]. These synthetic sensitivities continue to hinder a systematic understanding of how glass structure influences membrane performance.

Here, we developed a float glass–inspired method using a liquid gallium bath to fabricate thin, freestanding amorphous glassy ZIF-62 membranes ($a_g$ZIF-62 membranes). By matching surface energy between molten ZIF-62 and liquid gallium, this approach suppresses dewetting and enables uniform spreading of the viscous melt. To facilitate handling and testing, the resulting membranes were stamp transferred from the gallium bath onto porous polymer substrate. Unlike solid substrates that often cause delamination or rupture, the smooth, electron-rich gallium surface supports continuous

vitrification and yields defect-free membranes with tunable thickness. Beyond processing, we show that coordinatively unsaturated nitrogen sites in $a_g$ZIF-62 contribute directly to gas transport. Post-synthetic methylation with (TMS)CHN$_2$ selectively blocks these sites, leading to a significant drop in $CO_2$ permeance and a rise in activation energy, consistent with the loss of transient adsorption interactions. While recent simulations have suggested that a small fraction (0.8–1.4%) of coordination defects may form in glassy ZIFs, direct experimental evidence remains lacking. In this work, we not only capture such defects through post-synthetic methylation but also demonstrate their functional role in enabling facilitated $CO_2$ diffusion. Temperature-dependent permeation further supports a defect-assisted diffusion mechanism, where exposed N1 nitrogen atoms in ligands promote $CO_2$ diffusion via surface hopping. Furthermore, we uncover a second critical factor affecting membrane performance: the emergence of crystalline ZIF-zni impurities under specific precursor-mixing conditions. Although present in small amount, this minor phase significantly reduces gas sorption capacity and impairs membrane performance, highlighting the sensitivity of glass formation process.

Together, these findings establish a generalisable strategy for shaping glassy MOFs into membranes and offer mechanistic insight into the relationship between local disorder and selective gas transport. By coupling interfacial processing with chemical adjustment, this work demonstrates how structure and function can be simultaneously engineered in MOF glasses for membrane applications.

## Fabrication of agZIF-62 membrane on liquid metal bath

The reported procedure for the production of ZIF-62 crystal[4] (Fig. 1A) was adopted in this work. A single batch synthesis produces 7.4 g of phase-pure ZIF-62, exhibiting a crystallinity of over 99.7% as determined by Rietveld refinement of the powder XRD patterns (Fig. 1B). The linker ratio of the as-synthesised ZIF-62 was 1.00:4.39 (imidazole: benzimidazole), as calculated from the integration of their respective N1–H signals in the $^1$H NMR spectrum, which appear as well-separated peaks (Fig. 1C). The ZIF-62 crystals exhibit a particle size distribution of 5–10 μm. Differential scanning calorimetry (DSC) and heat capacity ($C_p$) measurements show the melting point ($T_m$) of 435 °C and glass transition temperature ($T_g$) of 325 °C (Fig. 1E).

Elimination of impurity phase from ZIF-62 is critical in forming high quality $a_g$ZIF-62 membranes. Formation of common impurity phase of ZIF-zni was suppressed by adjusting the order of mixing solution and the highest purity was found by the order of benzimidazole-imidazole followed by complete mixing and addition of zinc nitrate hexahydrate as demonstrated by X-ray diffraction (XRD) (Supplementary Fig. S1). To systematically assess the impact of precursor addition on phase formation, we investigated all six possible permutations for introducing the three precursors (labelled Protocols P1–P6, Table S1). In each protocol, components imidazole, benzimidazole, and zinc nitrate hexahydrate were added in a unique order under continuous stirring. Powder X-ray diffraction revealed that five protocols (P2-P6) yielded a secondary phase of ZIF-zni, whereas P1 produced a phase-pure ZIF-62. Detailed procedures and corresponding diffraction data for all six protocols are presented in Supplementary Fig. S1. Formation of low crystalline phase is related to the washing and drying steps. Methanol rinsing until DMF is completely removed, as confirmed from $^1$H NMR spectrum of the washing solution, followed by air oven drying, yields the highest crystallinity (99.7%) in ZIF-62.

For float glass synthesis of $a_g$ZIF-62 membranes, the high-purity ZIF-62 crystals were gently ground and pelletized using a hydraulic press operated at 50 MPa for 10 min. Piece of ZIF-62 crystal weighing 1–5 mg was taken from the pellet by reverse tweezer and placed on a home-made liquid metal bath contained on a glass side kept within an alumina crucible. A flow chart (Fig. 1D) outlines the procedures. Iridium was sputtered onto the glass slide to improve wetting between Gallium

 

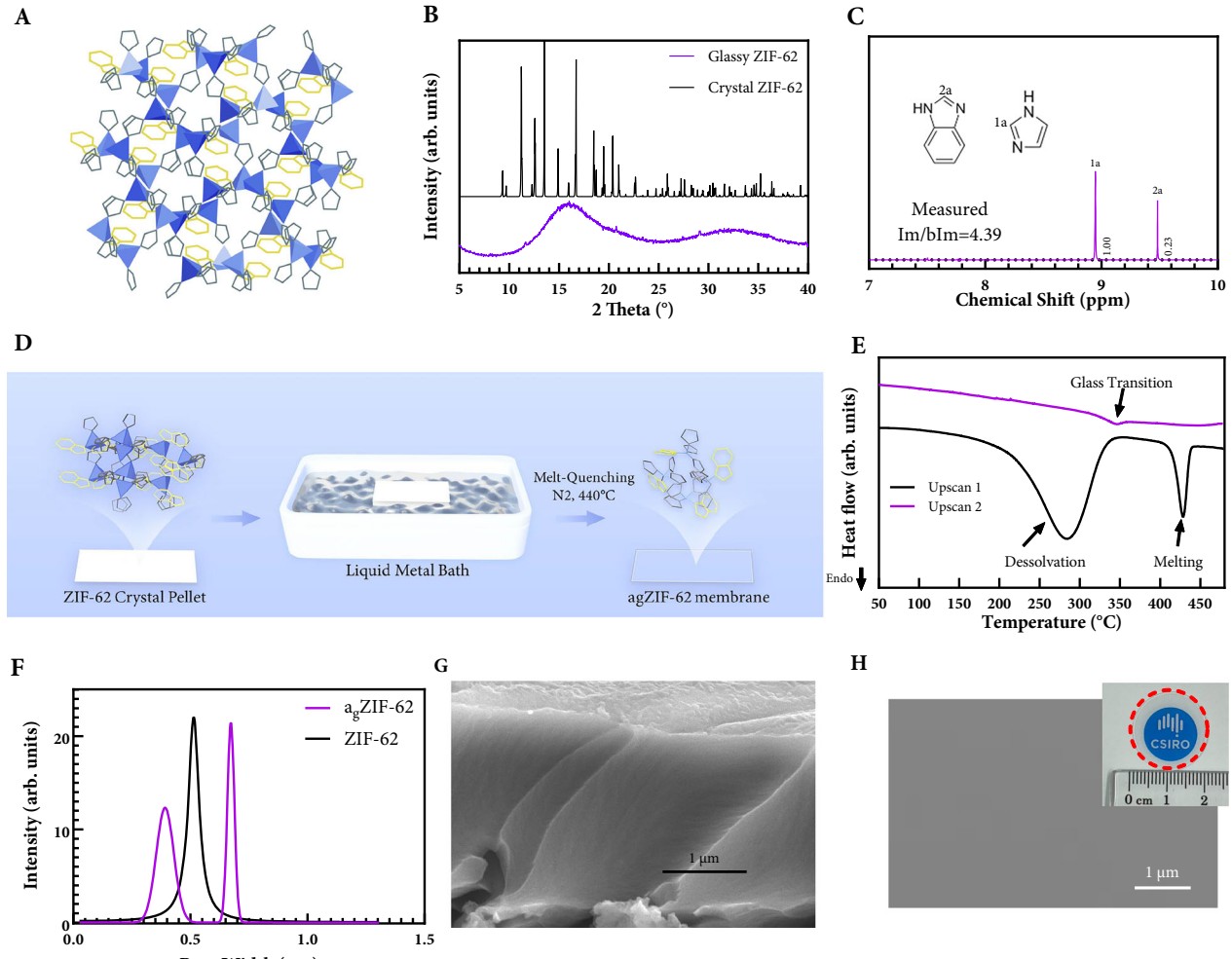

**Fig. 1 | Float glass fabrication of a_gZIF-62 membranes. A** Schematic of crystal structure of ZIF-62. **B** XRD pattern of ZIF-62 pellet and a_gZIF-62 membranes. (Blue: Zinc, Yellow: Benzimidazole, Black: Imidazole). **C** $^1$H NMR of a digested a_gZIF-62 membrane. **D** Flow chart of float glass fabrication method. ZIF-62 and the derived materials are shown in white. **E** DSC signal to determine $T_m$ of the ZIF-62 batch and cyclic scans with heating and cooling rates of 20 °C min$^{-1}$ to determine $T_g$. **F** Pore size distribution derived from PALS of ZIF-62 crystal and a_gZIF-62 membranes. SEM image of cross-section (**G**) and top-surface (**H**) of freestanding a_gZIF-62 membranes (red circle) with an insert of digital photograph.

bath and glass slide. Gallium (Ga, 99% Sigma Aldrich) was used as the liquid metal bath. Ultra-high purity $N_2$ was used as a protective atmosphere. A melting process followed by gas quenching was carried out for vitrification of ZIF-62 pellets into freestanding a_gZIF-62 membranes. A recent work that studied the effect of tempering of monolithic ZIF-62 glass found that extended thermal treatment further reduces the cavity size[4]. In our work, we used cold gas quenching and found an insignificant impact on gas sieving performance. Other low-toxicity liquid metals such as pure tin, gallium−tin (10%) and gallium-indium (25%) were also investigated. The a_gZIF-62 membranes can be formed on all these liquid metal bath, with slight differences in curvature and thickness which stems from different interfacial energy between molten ZIF and liquid metal bath. A table summarizing the liquid metals and alloys surface tension is provided in the Supplementary Table S2. The Ga bath was recycled between membrane making by removing gallium oxide layer with a plastic blade before placing ZIF-62 pellet (Methods). The previously reported bubble formation in glassy MOF is prevented by complete DMF removal. The a_gZIF-62 membranes were obtained from liquid metal bath after the melt-quenching process. Float glass processing was also effective in fabricating other glassy MOF membranes such as a_gZIF-62 (Co) (Supplementary Fig. S2), but their performance is not further investigated in this work. Mechanical robustness was not evaluated in this study and

may require further investigation in relation to surface structure and vitrification conditions. Excessive liquid metal was removed from the membranes by brushing with diluted HCl (0.07 wt%) followed by vacuum oven storage at 90 °C for complete drying. No detectable amount of Ga could be picked up from either side of the result membranes by EDS (Supplementary Fig. S3).

The a_gZIF-62 membranes prepared via float glass process were highly amorphous and porous, as confirmed by XRD and PALS. PALS analysis reveals a broader and bimodal pore size distribution in glassy ZIF-62 compared to the single, sharper peak observed in crystalline ZIF-62 (Fig. 1F). The glass exhibits one peak slightly smaller and another slightly larger than the crystalline pore size, suggesting the emergence of both constricted and expanded local environments upon vitrification. This bimodality reflects increased structural heterogeneity and is consistent with the collapse of medium-range order, in agreement with prior simulations and spectroscopic analysis of monolithic crystals[19]. The appearance of the larger pore component may also correspond to geometric motifs such as five-membered ring defects, as indicated by a recent computational study[25]. Surface area of a_gZIF-62 membranes by $CO_2$ adsorption is reduced (252 m$^2$/g) compared to its crystalline counterpart (316 m$^2$/g). The membranes show high levels of transparency and thinness, as characterised by a typical thickness of 2−10 μm corresponding to the weight of crystal ZIF-62

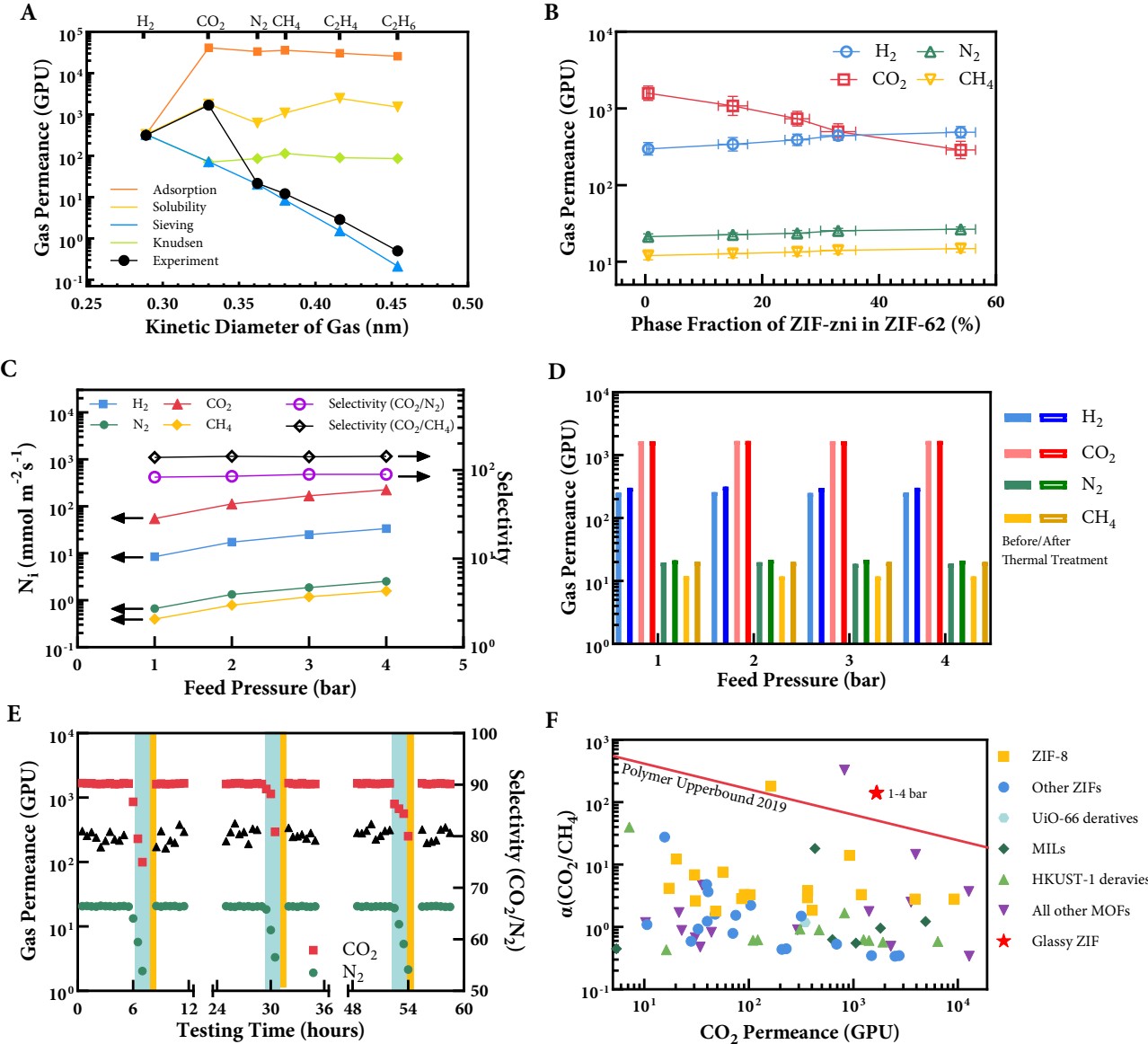

**Fig. 2 | Pure-gas permeation and stability of a_gZIF-62 membranes. A** Single gas permeance as a function of the gas kinetic diameter membrane at 298 K and 1 bar. **B** Effect of ZIF-zni phase on the gas permeance of result glassy MOF membranes. **C** Pressure-dependent gas flux. **D** Gas permeance before and after thermal treatment. **E** Three-day separation stability of the a_gZIF-62 membrane tested with humid feed (blue region) and helium drying (yellow region). **F** Comparison of the separation performances of the glass membrane with literature. Performance data of reported membranes are compiled from Qian et al.[46].

used for pellet making (Supplementary Fig. S4). The a_gZIF-62 membranes, thinner than 1 μm approach critical thickness for elasticity-to-plasticity transition[26], therefore, it is not evaluated as gas separation membrane in this work. The a_gZIF-62 membranes exhibit a refractive index of 1.55, which is in good agreement with both predicted and measured values[27]. Excellent homogeneity highlights the effectiveness of liquid metal bath in preparation for thin freestanding a_gZIF-62 membranes.

## Gas permeation, impurity, and agZIF-62 membrane stability

Gas diffusivities were measured for $CO_2$, $N_2$, $H_2$, $CH_4$, $C_2H_4$, $C_2H_6$, and $SF_6$ using an isometric method in a constant-volume–variable-pressure rig (Supplementary Fig. S5). The time-lag method was used for calculation of diffusion coefficients for each gas (Method). Elimination of grain boundary was confirmed by XRD and SEM analyses, preventing the "leaky" behaviour often observed in crystalline MOFs[28]. As a result,

the a_gZIF-62 membranes demonstrated typical molecular sieving characteristic: gas diffusivity is reduced by five orders of magnitude as the molecular diameter increases from 2.89 to 5.5 Å (Fig. 2A). This is further validated by computational modelling (Method). Modelling results confirms that $CO_2$ permeation is predominantly driven by adsorption-enhanced apparent solubility, while $H_2$ transport benefits from a minor Knudsen contribution due to its low molecular mass (Simulation Note in Supplementary Text). A sudden decrease in permeance between $C_2H_4$ and $C_2H_6$ suggests a cut-off diameter for a_gZIF-62 cage is between 0.42 and 0.45 nm, which agrees well with simulation[29]. Remarkable high selectivities of 83.3 for $CO_2/N_2$ and 138.9 for $CO_2/CH_4$ separation are achieved, exceeding the current performance limits of polymer and mixed matrix membranes[30]. Experimental data agree well with gas permeation model fitted using previously proposed method[31]. Notably, the $CO_2$ displayed an anomalously high permeation rate—surpassing that of $H_2$ despite its larger kinetic diameter—suggesting a distinct transport mechanism. A good agreement

was achieved, with $CO_2$ permeability primarily attributed to adsorption-enhanced apparent solubility, and $H_2$ permeability dominated by molecular diffusivity and minor Knudsen contribution. The Knudsen contribution is only significant for light gases such as $H_2$. Overall, pure-gas diffusivities are consistent with prior reports on $a_gZIF$-62 prepared by melt-quenching monolithic ZIF-62[4], and also thicker membranes (10–30 microns) fabricated by seeded solvothermal growth[11,24] and electrochemical deposition[12]. Notably, the float-glass process confers significantly improved uniformity and reduced thickness (2–10 micron), making it scalable for high-performance gas separations.

Our initial trials for glassy MOF membranes exhibited unexpected $H_2/CO_2$ selectivity, which was also observed in some recent studies[11,12]. This inconsistency was attributed to the presence of an impurity phase, ZIF-zni, which was commonly found in both commercial and some lab-synthesised ZIF-62 crystals prepared by solvothermal methods. By changing the order of addition, we identified inhomogeneous mixing of precursors as the prime source of ZIF-zni impurity (Supplementary Table S1). Specifically, adding benzimidazole before imidazole and then zinc nitrate produced a phase-pure membrane, whereas reversing the sequence led to up to mixture with 54% of ZIF-zni (Supplementary Fig. S1). DSC thermograms revealed that the impurity broadens the endothermic melting peak by -14 °C (Supplementary Fig. S6), probably due to restricted Zn–imidazole bond rotation compared with the benzimidazole counterpart. This observation suggests that ZIF-zni may reduce $CO_2$ permeance by limiting adsorption sites and disrupting the defect network, while having minimal impact on $H_2$ due to its size-selective diffusion. Although membranes with significant ZIF-zni retained comparable $H_2$ permeance (289.5 GPU), their $CO_2$ permeance is only 277.4 GPU, implying a different pathway in the pure $a_gZIF$-62 domains (Fig. 2B). By carefully controlling the precursor mixing to minimize ZIF-zni formation, we achieved consistently high $CO_2/H_2$ selectivity. These observations unify previous contradictory results by demonstrating that precursor mixing significantly impacts membrane phase purity and, ultimately, gas separation performance.

Stability of the $a_gZIF$-62 membranes was evaluated across multiple conditions, including feed-pressure variations (1 to 4 bar), long-term storage at high temperature (48 h at 200 °C), and moisture exposure with humid feed. In the tested feed pressure range, linear flux increment for $H_2$, $CO_2$, and $N_2$ (Fig. 2C) confirmed the total elimination of grain-boundary defects. Defective polycrystalline membrane usually shows decent gas separation properties at 1 bar, especially for $CO_2$, because $CO_2$ can condense at grain boundary defects and use it as additional transport pathway[32]. However, this pathway loses separation performance at increased transmembrane pressure[32]. In contrast, pressure-independent gas permeability showcases the structural integrity of glassy $a_gZIF$-62 membranes. The selectivity of $a_gZIF$-62 membranes, calculated from single-gas permeation tests at 1–4 bar, is independent of transmembrane pressure (Fig. 2C). This behaviour is typical of ideal membranes with rigid pores and stable channels[33], and is more commonly observed in inorganic materials like zeolites and carbon molecular sieves than in polymers[34]. The strong metal–ligand bonds in ZIF-62 likely underpin this stability, as predicted by previous modelling study[23]. Heat treatment in a vacuum oven at 200 °C for 48 h did not compromise gas-sieving performance (Fig. 2D), further highlighting the high thermal stabilities of the $a_gZIF$-62 membranes. A slight increase in gas permeance except $CO_2$ was observed, potentially due to the rearrangement of amorphous structure after high-temperature storage. The $CO_2$ permeance remains almost unchanged after thermal treatment, implies an additional contribution beyond molecular sieving. The thermal stability of agZIF-62 membrane in operation conditions agrees with minimal weight loss in inert atmosphere in TGA (Supplementary Fig. S7). In a three-day continuous permeance test, $a_gZIF$-62 membrane was first exposed to dry $CO_2$ or

$N_2$, followed by 1 h of humidified feed introduced via a water bubbler. Moisture was then purged away from the system using 50 mL/min dry He for 30 min before resuming gas permeation test. Brief exposure to humid environment reduced permeance for all tested gases, but a short-term purging fully restored membrane performance (Fig. 2E). Water sorption blocked the pores reversibly and kept the framework intact. Similar water stability has been observed for thicker $a_gZIF$-62 membranes as well[11]. Overall, these results underscore that $a_gZIF$-62 membranes exhibit robust performance in typical operating environments.

## Uncoordinated sites in agZIF-62 membrane facilitates CO2 transport

The relevance between $CO_2$ permeability and ZIF-zni impurity phase in glassy ZIF membrane motivates the investigation of $CO_2$ transport mechanism. Scission and renewal of the Zn–N bonds upon melting was speculated from molecular dynamic simulations[35] and ultrahigh-field $^{67}Zn$ NMR[36]. Nitrogen atoms of dangling imidazole linkers in ZIF-62 might form carboxyl groups during $^{13}CO_2$ sorption[20]. Therefore, the high $CO_2$ permeability observed in $a_gZIF$-62 membrane is hypothesized to arise from the presence of exposed nitrogen atom in the dissociated ligand, which act as $CO_2$ adsorption sites and facilitate $CO_2$ hopping. To verify this hypothesis, a chemical modification approach consisting of post-treatment of the membrane with (trimethylsilyl) diazomethane ((TMS)CHN$_2$) was employed (Fig. S8). The (TMS)CHN$_2$ was chosen due to its comparable shortest-projected-molecular-diameter (0.32 nm) to the pore windows of $a_gZIF$-62, therefore might methylate the accessible nitrogen sites at solvothermal conditions, potentially blocking them and altering $CO_2$ adsorption behaviour. Three groups of samples were prepared. The vitrified ZIF-62 membrane without (TMS)CHN$_2$ was assigned as Group 1. Group 2 was prepared by solvothermal treating ZIF-62 crystals with (TMS)CHN$_2$/hexane solution, followed by washing, pelletizing and vitrification. The Group 3 was prepared by vitrification of pelletized ZIF-62 and followed by (TMS)CHN$_2$ treatment and washing. Detailed experimental steps are available in supporting information (Supplementary Fig. S9).

To examine the structural consequences of vitrification and (TMS)CHN$_2$ treatment, porosity and ligand composition were analysed across membranes from Groups 1–3. Physisorption analysis confirmed that most of the porosity was preserved in all three groups (Fig. 3A). Group 3 exhibited a substantially lower specific surface area (191.3 m²/g) compared to Group 1 (287.1 m²/g) and Group 2 (277.6 m²/g), underscoring the pronounced impact of its (TMS)CHN$_2$ treatment condition. To probe the coordination environment, ligand composition was further analysed using ATR-FTIR for membrane samples and multipresat $^1H$ NMR after digesting each membrane in a (DCl/D$_2$O/DMSO-6d) solution (Method). In FTIR, the distinct peak at 2950 cm$^{-1}$ in Group 3 suggests C–H stretching was observed (Supplementary Fig. S9). In $^1H$ NMR, no signals other than those attributed to ZIF-62 were detected in Group 1. Group 2, which was treated with (TMS)CHN$_2$ prior to vitrification, showed very similar FTIR and $^1H$ NMR signals to Group 1. In contrast, Group 3, treated with (TMS)CHN$_2$ after vitrification, exhibited distinct signals at 3.64 ppm in $^1H$ NMR. These signals were assigned to 1-methylimidazole based on reference spectra and literature data[37]. In parallel, Raman spectra confirmed a reduction in the C–N stretching vibration (-1280 cm$^{-1}$) in Group 3 compared to Group 1, consistent with the presence of methylated imidazole species. Mapping of the 1280/1600 cm$^{-1}$ intensity ratio further revealed a uniform distribution of this chemical change across the membrane surface, serving as a spatial indicator of successful and homogeneous post-synthetic modification (Supplementary Fig. S13). Integrated signal area in $^1H$ NMR suggests that concentration of 1-methylimidazole is 0.8% of the total ligand presented in the methylated $a_gZIF$-62 membrane (Group 3). Notably, this value aligns with the predicted coordination defect

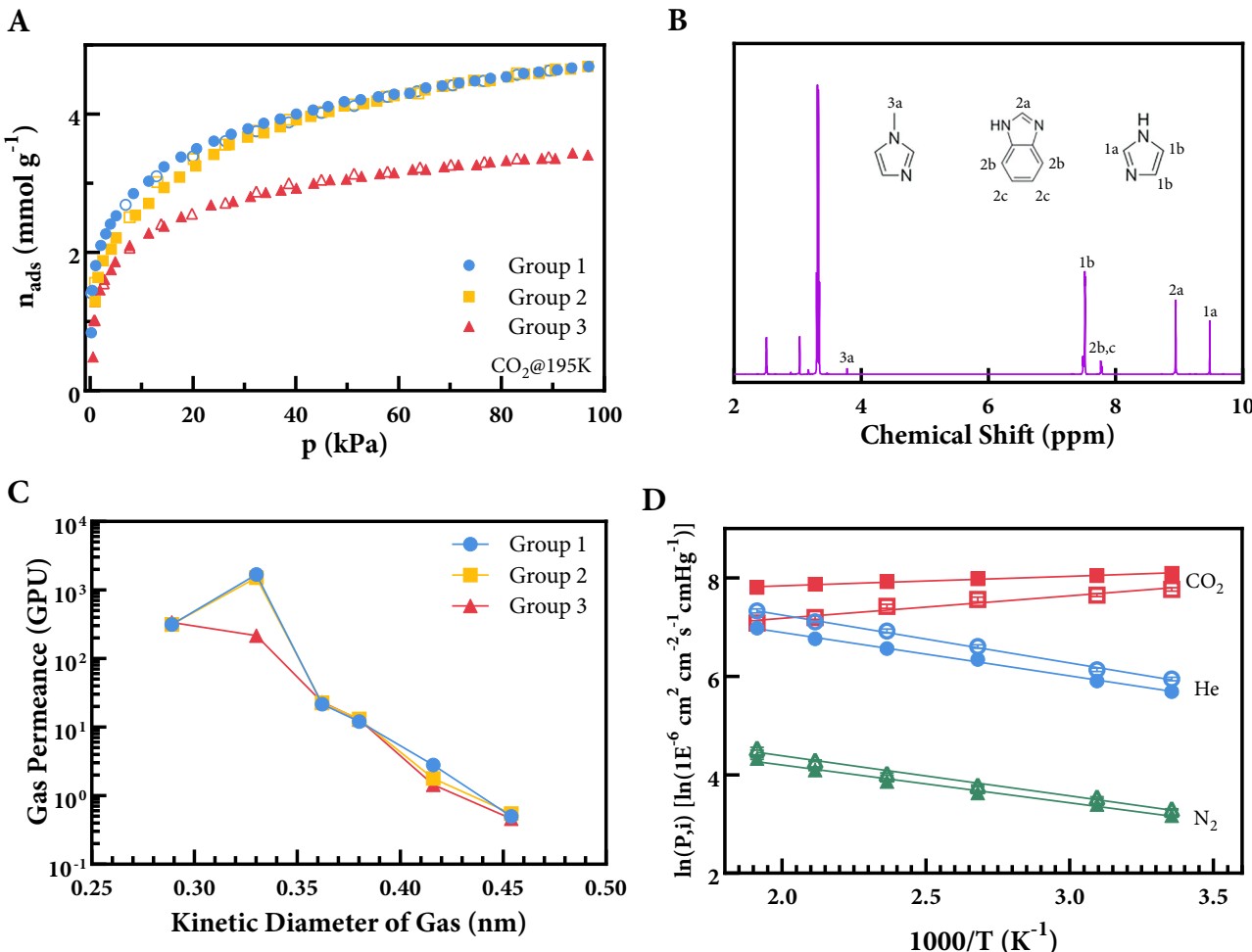

**Fig. 3 | Post-synthesis modification of $a_gZIF$-62 membranes and its effect on $CO_2$ transport. A** Schematic of solvothermal $(TMS)CHN_2$ treatment of Group 1 (unmodified $a_gZIF$-62), Group 2 (methylation then vitrification) and Group 3 (vitrification then methylation). **B** $^1H$ NMR spectrum of digested Group 3 sample.

**C** Single gas permeance as a function of the gas kinetic diameter for Group 1–3 membranes at 298 K and 1 bar. **D** Arrhenius plot for gas flux with respect to 1000/T for Group 1(solid) and Group 3 (hollow).

level (0.8–1.4%) in amorphous ZIF-62 obtained from machine learning–guided HRMC simulations[25]. In the same study, a structural analysis revealed the presence of ~1.4% three-membered ring defects and ~0.8% five-membered ring defects in glassy ZIF-62. These five-membered ring defects may correspond to broken Zn–N linkages, offering a plausible structural origin for the experimentally methylated uncoordinated nitrogen sites. We note that interfacing with the liquid metal bath may contribute to coordination defect formation during vitrification, although this remains unconfirmed and requires further spatially resolved analysis. This structural motif is consistent with PALS results (Fig. 2F), which suggest the presence of local free-volume elements that could accommodate such irregular geometries. This agreement between experimental observation and simulation not only supports the existence of such defects but also provides mechanistic insight into their probable geometric nature.

The presence of this methylated species suggests that $(TMS)CHN_2$ selectively reacted with a small fraction of N1 nitrogen atoms on imidazole ligands in the vitrified membrane. The key difference between Group 2 and Group 3 lies in the sequence of treatment: in Group 2, exposure to $(TMS)CHN_2$ prior to vitrification resulted in no detectable methylation. However, in Group 3, post-vitrification treatment enabled the reaction. Unsuccessful methylation of the crystalline ZIF-62 suggests that the imidazole nitrogen atoms were fully coordinated, which is supported by the high degree of crystallinity (>99%) of as-prepared ZIF-

62 (Fig. 1B). By contrast, the successful reaction in Group 3 under the same solvothermal condition suggests that vitrification exposed some previously coordinated nitrogen sites, as speculated in previous studies[20,35,38]. Moreover, 1-methylimidazole does not coordinate with zinc ions, as its N1 position is occupied by a methyl group, preventing lone electron pair donation[39]. Thus, the formation of 1-methylimidazole in Group 3 indicates that these nitrogen atoms were no longer coordinated at the time of methylation. This experimental evidence aligns with previous theoretical studies[38], which suggest that vitrification disrupts the original coordination structure of crystalline ZIF-62, breaking small portion of imidazole–zinc bonds and generating uncoordinated sites. The subsequent methylation of exposed nitrogen sites further supports the existence of such defects in the glassy framework.

We further investigated the relationship between uncoordinated sites in $a_gZIF$-62 membrane and $CO_2$ diffusion through them. Membrane samples with similar thickness from Groups 1–3 were tested in home-made isometric gas permeation rig and temperature-controlled isobaric rig (Supplementary Fig. S11). Single gas permeation measurements were conducted in isometric rig using pure gas ($H_2$, $CO_2$, $N_2$, $CH_4$, $C_2H_4$, $C_2H_6$, $SF_6$) at 25 °C under a feed pressure of 1 bar. Figure 3C shows the results for three groups of samples—Group 1 (blue circles), Group 2 (yellow squares), and Group 3 (red triangles). The similarity in permeance-kinetic diameter across all groups suggests that the dominant transport mechanism remains molecular sieving. Notably,

Group 1 and 2 present highest permeance for $CO_2$ over all tested gases, which is consistent with untreated $a_g$ZIF-62 membranes. Interestingly, in Group 3, where uncoordinated sites were plugged by methylation from (TMS)CHN$_2$ treatment, the reversed $CO_2/H_2$ selectivity was 'corrected' to $H_2/CO_2$ selective performance, suggesting the loss of additional diffusion pathway for $CO_2$. After methylation, the $a_g$ZIF-62 membrane shows more typical molecular sieving behaviour where pore size and penetrant geometry dominate gas permeation. This is further validated by the pure $C_2H_4$ permeation through Group 1–3 (Fig. 3C). The planar and rigid structure of $C_2H_4$ makes it more sensitive to the geometry and connectivity of membrane pore channels[40]. The comparatively lower $C_2H_4$ permeance for Group 3 suggests that methylation induces a more distorted channel. In membranes containing ZIF-zni impurity (Fig. 2B), the more rigid and tightly coordinated framework may inhibit defect generation during vitrification—potentially falling below the percolation threshold required for facilitated $CO_2$ transport to emerge[11,24]. Taken together, these results suggest that the uncoordinated sites generated from vitrification process contributes to the fast $CO_2$ transport in the $a_g$ZIF-62 membrane via facilitated transport, in additional to pore flow.

The observed decrease in $CO_2$ permeance and the reversal of selectivity following post-synthetic methylation is consistent with our hypothesis that uncoordinated sites formed during vitrification facilitates $CO_2$ transport. Specifically, these uncoordinated nitrogen sites may act as transient sorption centres that enable hopping-type surface diffusion[41–43], complementing molecular sieving. However, we acknowledge that other mechanisms may also contribute to the observed behaviour. For example, methylation may induce subtle distortions in the pore network or reduce the flexibility of the framework, either of which could influence gas permeation profiles. Despite the small portion of methylated imidazole (-0.8%), the pronounced change in $CO_2$ selectivity suggests that the defect-blocking hypothesis is a plausible dominant factor as well, especially given the minimal impact on other gases (Fig. 3C). Nevertheless, we emphasize that the precise contribution of uncoordinated sites remains to be fully elucidated. Further studies, such as molecular simulations and sorption experiments targeting uncoordinated sites, may help disentangle these effects and establish the generality of this transport behaviour in glassy MOF membranes.

To shed light on the contribution of facilitated transport to the overall $CO_2$ permeation, we conducted temperature-dependent (25–250 °C) single gas permeation test for Group 1 and 3 in an oven-based isobaric gas permeation rig (Fig. 3D). Helium (0.26 nm) was used as representative molecule for $H_2$ (0.289 nm) due to their similar molecular size and the safety considerations[44]. Helium is smaller than $H_2$ but twice as heavy; therefore, it is expected to diffuse more slowly than $H_2$. Activation energy for Helium permeation calculated from Arrhenius plot are very similar for Group 1 and 3, with fitted value of 7.34 and 7.97 kJ/mol, respectively. Helium transport through Group 3 sample was slightly more activated, suggesting a more disrupted pore channel system induced by methylation. Methylation of the uncoordinated sites in Group 3 diminishes the facilitated diffusion of $CO_2$, as indicated by a significant loss of activation energy from −1.59 kJ/mol for Group 1 to −3.89 kJ/mol for Group 3. Regardless of the contribution from uncoordinated sites, the $CO_2$ permeation still shows negative activation energy, which is related to the high polarity of $CO_2$ molecule and the excessive amount of imidazole species in the framework. The difference in $CO_2$ permeation activation energy between Groups 1 and 3 suggests that uncoordinated sites in $a_g$ZIF-62 membrane significantly enhance $CO_2$ transport, increasing it nearly five-fold.

In summary, our results suggest that uncoordinated sites formed during the vitrification of ZIF-62 membranes play a central role in enhancing $CO_2$ transport beyond molecular sieving. We demonstrated that methylation of a small fraction of these sites causes a pronounced

drop in $CO_2$ permeance and reverses $CO_2/H_2$ selectivity. These findings highlight a broader conceptual insight: disorder-induced local environments in amorphous frameworks can facilitate selective gas diffusion. Similar phenomena have been reported in zeolites, where intrinsic framework defects—such as missing linkers or silanol nests—serve as low-energy sorption sites, enabling sorption-enhanced transport for polar molecules[42,43]. In this context, the defect-rich nature of glassy MOFs may offer comparable diffusion advantages, provided the local chemistry remains favourable after vitrification. This analogy underscores the potential of uncoordinated defect as a tunable structural feature that can be leveraged to optimize separation performance in non-crystalline porous materials.

Beyond ZIF-62, the applicability of this interface-assisted vitrification strategy to other MOF glasses remains an open question. In particular, the selection of liquid metal for each MOF system would require further study of melt–bath interfacial properties, including surface tension, wettability, and chemical compatibility, which are not addressed in this work.

## Discussion

Overall, we have demonstrated the promise of using float glass-inspired liquid metal bath for fabricating homogeneous glassy MOF membranes with thickness down to two microns, together with well-preserved porosity that achieved excellent molecular sieving. The proposed method echoes with very recent calls for liquid-processable, free-standing MOF glass membranes that made from 100% bulk ZIFs and tackle important and highly challenging gas separation[4,40]. Moreover, we found that the ultrafast $CO_2$ permeation through glassy ZIF-62 membrane was arising from the phase impurity and uncoordinated sites generated during vitrification. Post-synthetic methylation allows us to fine-tune these defects and consequently provides evidence for sorption-facilitated $CO_2$ diffusion, as observed experimentally in this work and in recent studies[12,45]. These findings highlight how interfacial control and defect engineering can be combined to achieve processable, high-performance glassy MOF membranes.

## Method

### Materials

Zinc nitrate hexahydrate (98%, Sigma-Aldrich), cobalt(II) nitrate hexahydrate (99%, Sigma-Aldrich), benzimidazole (98%, Sigma-Aldrich), imidazole (99.5%, Sigma-Aldrich), dimethylformamide (99.8%, Merck), dichloromethane (99.8%, Merck), methanol (99.9%, Merck), hydrochloric acid (35 wt%, Merck), deionized water (15.2 MΩ·cm, Milli-Q system), gallium (99%, Sigma-Aldrich), tin (99%, Sigma-Aldrich), gallium–tin alloy (10%, Sigma-Aldrich), gallium–indium alloy (25%, Sigma-Aldrich).

### Synthesis of ZIF-62 (Zn and Co)

Phase-pure ZIF-62 was synthesised using a modified solvothermal method with improved precursor addition and purification steps. For Zn–ZIF-62, 3.20 g benzimidazole (bIm) and 9.51 g imidazole (Im) were sequentially dissolved in 240 mL DMF, followed by the addition of 4.98 g zinc nitrate, yielding a Zn:Im:bIm ratio of -3:25:5. Each precursor was stirred for at least 5 min before the next was added. This specific order, combined with intermediate stirring, was critical for suppressing ZIF-zni formation. The solution was heated at 130 °C for 72 h in a sealed Teflon-lined jar. Crystals were isolated by centrifugation, washed with DMF and methanol, and dried at 45 °C before vacuum activation (150 °C, 72 h). A total of 7.35 g product was recovered from the mother liquor. ZIF-62(Co) was prepared similarly using adjusted precursor ratios.

### Fabrication of agZIF-62 membrane

High-purity ZIF-62 crystals were gently ground and pelletized at 50 MPa for 10 min. Pelletized fragments (1–5 mg) were transferred

from the pellet onto a home-made liquid metal bath (Ga, 99%, Sigma-Aldrich) supported on an iridium-coated glass slide within an alumina crucible. The crucible was placed under a flowing UHP $N_2$ tube furnace, and a melt-quenching process was used to vitrify ZIF-62 into free-standing $a_g$ZIF-62 membranes. ZIF-62 pellet was heated up to 450 °C at a rate of 10 °C min$^{-1}$ and then held at this temperature for 5 min, after which it was cooled down under cold UHP $N_2$ flowing at 300 ml min$^{-1}$. Other low-toxicity liquid metals (Sn, Ga–Sn, Ga–Ir) might be used as well. The Ga bath was recycled between batches by scraping off the oxide layer with a doctor blade. After quenching, the $a_g$ZIF-62 membrane could be removed from hot gallium bath or stamp transferred onto glass slide, carbon tape, or porous PTFE filter for characterisation or gas permeation test. Excess liquid metal on glassy MOF was removed by brushing with 0.07 wt% HCl, followed by overnight drying in vacuum oven at 90 °C.

### Characterisation

The cross-sectional morphology and thickness of agZIF-62 membranes were examined using field emission scanning electron microscopy (FE-SEM, Zeiss Merlin, Germany). Crystalline and amorphous phases were identified by X-ray diffraction (XRD, Bruker D8 Focus, Germany) with Cu-Kα radiation (λ = 1.54 Å). Particle size of as-prepared ZIF-62 crystal was determined by laser diffraction (Saturn II Particle Size Analyser, Micromeritics, USA). Pore size distribution was assessed by positron annihilation lifetime spectroscopy (PALS, EG&G Ortec, USA), and specific surface areas were calculated using BET analysis of $CO_2$ isotherms (Micromeritics ASAP2420, USA). Refractive index measurements were performed by refractometry (RM 50, Mettler Toledo, Switzerland/USA). Molecular structure was analysed by ATR-FTIR (Nicolet 6700, Thermo Fisher, USA), and ligand composition was confirmed via $^1$H NMR spectroscopy (Avance Neo 500 MHz, Bruker, Germany) with powder sample digested using previously reported method[21]. Thermal stability and transitions were characterised by TGA, DSC, and cyclic $C_p$ measurements using a thermal analysis system (STAR System, Mettler Toledo, Switzerland/USA), with $T_g$ identified from the onset of the glass transition signal.

### Gas permeance measurements

The gas separation performances of agZIF-62 membranes were tested at 25 °C using a constant-volume/variable-pressure method. The downstream pressure was measured using a transducer down to $1.5 \times 10^{-6}$ torr, and the rate of pressure change (dp/dt) of pseudo-steady state was selected to calculate the permeance (J). Pure gas was tested in the sequence of $SF_6$, $C_2H_6$, $C_2H_4$, $N_2$, $CH_4$, $CO_2$, and $H_2$ at 1–4 bars. At least three replicates were tested, and the deviation was less than 5%. Permeance was calculated using the following Eq. 1:

$$J = 10^6 \times \frac{V_d}{P_{feed} \cdot T \cdot R \cdot A} \times \frac{dp}{dt} \qquad (1)$$

Where J is the permeance (GPU), 1 GPU = $10^{-6}$ cm$^3$(STP)/(cm$^2$·s·cmHg). $V_d$ is the calibrated permeate volume (cm$^3$), $p_{up}$ is the upstream pressure (cmHg), $A$ is the effective membrane area, $T$ is the operating temperature (K), $R$ is the gas constant (0.278 cmHgcm$^{-3}$(STP)K$^{-1}$) and dp/dt is the pseudo-steady state downstream pressure increase rate (cmHgs$^{-1}$).

The ideal selectivity ($\alpha_{x/y}$) for components x and y was defined as the ratio of gas permeability of the two components via Eq. 2.

$$\alpha_{\frac{x}{y}} = \frac{P_x}{P_y} \qquad (2)$$

The diffusion coefficient D was calculated from the time-lag method using Eq. 3:

$$D = l^2 / 6\theta \qquad (3)$$

Where $\theta$ denotes time lag.

### Modelling

To further elucidate the observed gas separation behaviour, a mathematical model was developed to describe the relationship between permeability, apparent solubility, and diffusivity:

$$P = S \cdot D \qquad (4)$$

Apparent solubility $S$ was expressed as a combination of adsorption affinity and gas–framework interaction energy:

$$S = \exp\left(\frac{\alpha\varepsilon}{RT} + \beta C_{Ads}(\sigma, \varepsilon, d, T)\right) \qquad (5)$$

where $\varepsilon$ is the potential energy (well depth), σ is the kinetic diameter, $d$ is the average pore size, $T$ is the temperature, and $C$Ads is the adsorbed-phase concentration. Adsorption was treated using the Topologically Integrated Mathematical Thermodynamic Adsorption Model, which analytically incorporates pore geometry and energy distribution.

Diffusivity $D$ was described as:

$$D = \exp\left(-\frac{\gamma\sigma^2}{RT}\right) + \delta\sqrt{\frac{8RT}{\pi M}} \qquad (6)$$

where $m$ is the molecular mass, and the two terms represent activated molecular sieving and Knudsen transport, respectively. The model was fitted to experimental data using a non-linear least-squares algorithm (lsqcurvefit) in MATLAB R2020a. Full derivations and fitting details are provided in Supplementary Note 1.

## Data availability

The data generated in this study are provided in the Supplementary Information and Source Data file. No custom code was generated. Source data are provided with this paper.

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

## Acknowledgements

This work was supported by CSIRO Manufacturing (internal funding to Z.X.). The authors acknowledge the Materials Characterisation Group of CSIRO Manufacturing, with special thanks to Dr Jo Cosgriff for NMR training, Dr Aaron Seeber and Dr Sherman Wong for XRD analysis, Dr Malisja de Vries and Mr. Mark Greaves for SEM training, and Dr Yesim Gozukara for DSC training. X.J. acknowledges helpful discussions with Dr Hamidreza Mahdavi (Monash University). The authors also thank A/Prof. Farshid Pahlevani (University of New South Wales) for high temperature characterisation.

## Author contributions

X.J. and Z.X. conceived the project and designed the experiments. X.J., X.W., and D.A. performed synthesis and characterisation experiments. D.N. assisted in membrane fabrication and testing. A.W.T. conducted mathematical modelling. H.W. provided input on MOF synthesis and analysis. Z.X. supervised the project. X.J. and Z.X. wrote the manuscript with contributions from all authors. All authors analysed the data, discussed the results, and approved the final version of the manuscript.

## Competing interests

The authors declare no competing interests.
