## [Transparent Peer Review file · Nature Communications]

Liquid Metal Interface Enables Glassy MOF Membranes with Defect-Mediated CO₂ Transport

Corresponding Author: Professor Zongli Xie

Version 0:

Reviewer comments:

Reviewer #1

(Remarks to the Author)

This manuscript reports a MOF glass membrane fabricated via a liquid metal method, adapted from techniques commonly used in traditional thin-film glass fabrication. The resulting membrane demonstrates impressive CO₂ transport performance and high selectivity over bulkier gas molecules. Overall, the paper offers valuable and interesting insights into membrane formation using MOFs. However, several aspects require further clarification and improvement before it can be considered for publication.

- 1 The main argument of this paper is to make mechanically robust membranes for separation. The authors claim the membrane surface is ultra smooth - but little experimental data was presented. This should be addressed, and how it would affect the mechanical properties?
2. The coating of metal on glass membrane is an important step to obtain the thin membrane, through regulating the surface tension. Would this coating impose any change of the MOF glass structure, and potentially block / facilitate gas transport?
3. The BET results of the MOF glass and crystal are higher than the literature values, please discuss the possible reason.
4. Another comment is on the ratio between imidazolate and benzimidazolate ligands - according to the NMR results, it seems the blm is more dominating phase - which is inconsistent with the literatures where the higher ratio of blm can lead to the formation of ZIF-7.
5. How would the interfacing with metal liquid can affect the bulk glass defects formation?
6. Dsc requires noting of the end/exo thermic direction

Reviewer #2

(Remarks to the Author)

The manuscript presents a liquid metal-assisted method for the preparation of high-quality glass MOF membranes. The anomalous CO₂ permeance of the glass ZIF-62 membranes reported in literature is also explored. Overall, this report is very interesting, and I believe it will attract the attention of researchers working on glassy MOFs. However, the writing of this manuscript is not sufficiently clear and intuitive. Some details are not described thoroughly, and there are still some basic errors (such as the missing of some figures, typos, format issues...). My main comments are:

1. I expect the authors to provide more experimental data on the vitrification of ZIF-62 membranes on different liquid metals, such as SEM/optical images of the prepared glassy ZIF-62 prepared on different liquid metal substrates. Also, I expect a quantitative explanation of the relationship between the surface tension of the liquid metals and the quality of the resulting glassy ZIF-62 membranes.
2. Is the proposed method applicable for other glass MOFs (e.g., ZIF-4 & TIF-4)? As the surface energy of the melted ZIF-4 or other MOFs may be different from that of ZIF-62, which liquid metal could be the best for the preparation of high-quality glass MOF membranes? Why?
3. Can the authors provide detailed information on determining the proportion of the zni impurities?
4. I really appreciate the methylation experiments presented in the work. Can the authors provide more detailed information on the methylated glass ZIF-62 (e.g., Raman mapping? Pore size analysis?)?

Additional points

1. Line 124: The authors reported that the proportion of the generated ZIF-zni impurity varies from the order in which different solutions are added during the solvothermal synthesis. It is relatively easy to understand the differences between P1 and P6. However, I am confused on the differences between P1 and P2. As suggested in Table S1, there is a considerable ZIF-zni proportion difference between the samples obtained from P1 and P2. How could it be?
2. Line 171-172. The authors claimed that the surface area of the glass ZIF-62 membranes by N2 is 252 m²/g. As glass ZIF-62 generally shows very little adsorption of N2, I am curious on what kind of test the authors used for the result?
3. The authors tried different liquid metals, e.g., gallium, pure tin, gallium-tin (10%), and gallium iridium (25%) for the synthesis of glass ZIF-62 membranes. Do these liquid metals affect the quality of the fabricated membranes? Gallium bath was selected for the following experiments. Why? Detailed material characterizations of the membranes are recommended.
4. As stated by the authors, the thickness of the prepared glass ZIF-62 membranes ranges from 2 μm to tens of micrometers, which is controlled by the quantity of the mother precursor, i.e., crystalline ZIF-62 powders. What are the thicknesses of the crystalline ZIF-62 pellets? Does the thickness of the ZIF-62 pellet influence the thickness of aZIF-62 membrane? Also, is the size of the prepared aZIF-62 membrane the same as the mother crystalline ZIF-62 pellet?
5. The authors observed that the generated ZIF-zni impurity show negligible effects on the H₂ permeance while has an adverse effect on the CO₂ permeance. Do the authors have any assumptions/proof on this result?
6. Line 276: The authors argued that the methylation of the uncoordinated sites of aZIF-62 membranes leads to considerably decreased CO₂ permeance, while it has very little influence on H₂/N₂/CH₄/C₂H₆ (Figure 3C). How could the H₂/N₂/CH₄/C₂H₆ permeance remain after the methylation process, given that this process introduces a decreased surface area of the aZIF-62 membranes as mentioned by the authors (Line 276).
7. It seems the C₂H₄ permeance varies from the Group 1-3 samples. The authors claimed that the methylation has negligible effects on the crystalline ZIF-62 powders since they have few uncoordinated sites. However, why is there a C₂H₄ permeance change in the Group 2 samples compared to the Group 1 samples?
Some minor issues are:
A table summarizing the surface energies of the used liquid metals is recommended.
The unit of the permeance in Table S2 is missing.
The lettering used for figure numbers in the article is sometimes uppercase and sometimes lowercase. Please make it consistent. Line 165. Figure 2f.
The font of the figure legends should be modified.
The details in determining the degree of crystallinity of the synthesized ZIF-62 powders should be given.
Line 191: Where is Supplementary Note 1?
Line 292: The authors mentioned "This structural motif is consistent with PALS results (Figure 2F) " The attached Figure 2F doesn't show any data on PALS results. Please correct this. Is this Figure 1F?
Line 275, where's Figure 3X? This figure is missing.

Reviewer #3

(Remarks to the Author)

Version 1:

Reviewer comments:

Reviewer #1

(Remarks to the Author)

The authors have addressed the comments raised in the last round of review. The paper now can be accepted for publication.

Reviewer #2

(Remarks to the Author)

Although certain questions remain, these can reasonably be addressed through future investigations. I therefore consider the manuscript suitable for publication in Nature Communications in its current form.

Reviewer #3

(Remarks to the Author)

Responses to the Reviewers' Comments for Manuscript NCOMMS-25-40107A

Reviewer #1 (Remarks to the Author):

This manuscript reports a MOF glass membrane fabricated via a liquid metal method, adapted from techniques commonly used in traditional thin-film glass fabrication. The resulting membrane demonstrates impressive CO₂ transport performance and high selectivity over bulkier gas molecules. Overall, the paper offers valuable and interesting insights into membrane formation using MOFs. However, several aspects require further clarification and improvement before it can be considered for publication.

Comment 1. The main argument of this paper is to make mechanically robust membranes for separation. The authors claim the membrane surface is ultra smooth - but little experimental data was presented. This should be addressed, and how it would affect the mechanical properties?

Response: We thank the reviewer for raising this point. We acknowledge that the phrasing in the original manuscript may have caused unintended confusion. Specifically, the term “robust” in our manuscript refers to the membrane's performance stability under elevated temperature and humid operating conditions (e.g., thermal treatment at 200 °C for 48 h, long-term permeation under humid feed), rather than mechanical robustness. We have revised the text accordingly to clarify this distinction and avoid misinterpretation.

Likewise, the description of “ultraflat” or “ultra-smooth” referred to the liquid gallium substrate, which provides a highly planar and chemically inert interface that facilitates uniform spreading of the molten ZIF-62 during vitrification. We did not intend to claim that the resulting membranes are atomically smooth or possess ultra-smooth surfaces in the mechanical sense. To further clarify this point, we have updated the relevant wording in the revised manuscript (Page 5).

To further improve the clarity, we also added the following sentence in the revised manuscript:

“Mechanical robustness was not evaluated in this study and may require further investigation in relation to surface structure and vitrification conditions.”

Comment 2. The coating of metal on glass membrane is an important step to obtain the thin membrane, through regulating the surface tension. Would this coating impose any change of the MOF glass structure and potentially block / facilitate gas transport?

Response: We appreciate the reviewer's thoughtful question. In our experiments, surface elemental analysis using EDX was conducted on both sides of the agZIF-62 membranes. No gallium-related signals were detected, which suggests that surface-level contamination is minimal.

To assess the practical relevance of any possible structural influence, we compared our membrane performance with values reported in the literature for glassy ZIF-62 membranes fabricated via other methods, including those with significantly greater thickness (e.g., 10–30 μm). Despite the differences in fabrication route and membrane geometry, the gas permeabilities, particularly for weakly interacting gases such as H₂ and N₂, are remarkably consistent across studies. This observation suggests that the key transport characteristics of agZIF-62 membranes are preserved, and that any

influence from the liquid metal interface, if present, does not appreciably alter the bulk gas transport behaviour. Our current data indicate that the gallium bath does not measurably impact the separation performance. A brief clarification has been added in the revised manuscript to reflect this interpretation: “We note that interfacing with the liquid metal bath may contribute to coordination defect formation during vitrification, although this remains unconfirmed and requires further spatially resolved analysis.”

However, we acknowledge that this does not rule out the possibility of gallium atoms being trapped within the microporous framework, such as in the cage interiors of agZIF-62. Determining the presence and spatial distribution of such internalised species would require high-resolution synchrotron-based techniques, which are beyond the scope of the current study. We consider this an interesting direction for future work.

Comment 3. The BET results of the MOF glass and crystal are higher than the literature values, please discuss the possible reason.

Response: We thank the reviewer for highlighting this point. We would like to clarify that the reported surface area of 252 m²/g was obtained via CO₂ adsorption at 273 K, not N₂ at 77 K. This was mistakenly labelled in the original manuscript and has been corrected in the revised version.

When using CO₂ at 273 K, values around 250 m²/g have also been reported in the literature for ZIF-62 glass samples with comparable activation conditions. For instance, Frentzel-Beyme et al. (Nat. Commun. 2022, DOI:10.1038/s41467-022-35469-3) reported values up to 246 m²/g. Our results are therefore consistent with prior findings. A brief clarification has been added in the revised manuscript (Page 5).

For clarity, we also modified the sentence in the revised manuscript:

“Surface area of agZIF-62 membranes by CO₂ adsorption is reduced (252 m²/g) compared to its crystalline counterpart (316 m²/g).”

Comment 4. Another comment is on the ratio between imidazolate and benzimidazolate ligands - according to the NMR results, it seems the bIm is more dominating phase - which is inconsistent with the literatures where the higher ratio of bIm can lead to the formation of ZIF-7.

Response: We thank the reviewer for pointing out this important inconsistency and apologise for our mistake. Upon re-examination, we confirmed that the ligand ratio stated in the original manuscript was incorrectly labelled. The ¹H NMR spectrum presented in Figure 1C is correct and clearly shows two distinct peaks corresponding to the N1-H protons of imidazole and benzimidazole, respectively. However, the assignment of the integration values was mistakenly reversed in the original text. The actual ratio, derived from the integral areas, is approximately 4.39:1.00 imidazole to benzimidazole, which is consistent with the expected composition of ZIF-62 reported in literature. We have corrected this error in the revised manuscript and clarified that imidazole is the dominant linker in our samples.

We appreciate the reviewer’s thorough reading, which helped us identify and rectify this misstatement. We have attached a revised version of Figure 1C below.

Comment 5. How would the interfacing with metal liquid can affect the bulk glass defects formation?

Response: We thank the reviewer for this important and thoughtful question. Given the membrane thinness (2–5 μm), the interfacial region is considerable and in contact with the liquid metal during vitrification, and this could possibly influence local coordination environments. This possibility is supported by well-established findings in float glass manufacturing where molten tin diffuses into the glass surface and induces localized structural defects near the interface (Johnson et al., *Hyperfine Interactions*, 1995, DOI: 10.1007/BF02146304; Colombin et al., *Journal of Non-Crystalline Solids*, 1977, DOI: 10.1016/0022-3093(77)90048-5).

As analogy, interfacial interactions with gallium could influence vitrification by localized heat dissipation or limited atomic diffusion. However, as noted in Supplementary Figure S3, we do not observe any detectable gallium signal via SEM-EDS at the membrane surface, suggesting that any such diffusion, if happened, is below the sensitivity of this technique. This limits our ability to experimentally resolve the influence of the metal interface on bulk defect formation at this stage.

We therefore consider this an open mechanistic question. Further spatially resolved spectroscopic methods or synchrotron-based technique will be necessary to determine whether and how the metal-MOF interface affects defect formation in the bulk glass region.

Comment 6. Dsc requires noting of the end/exo thermic direction

Response: Following the reviewer's suggestion, the DSC plots have now been updated to clearly indicate the endothermic and exothermic directions. The updated figure has been included in the revised manuscript (Figure 1E and Figure S6), with a note clarifying the thermal event direction has also been added to the y-axis for improved clarity. We have attached a revised version of Figure 1E below.

Reviewer #2 (Remarks to the Author):

The manuscript presents a liquid metal-assisted method for the preparation of high-quality glassy MOF membranes. The anomalous CO₂ permeance of the glassy ZIF-62 membranes reported in literature is also explored. Overall, this report is very interesting, and I believe it will attract the attention of researchers working on glassy MOFs. However, the writing of this manuscript is not sufficiently clear and intuitive. Some details are not described thoroughly, and there are still some basic errors (such as the missing of some figures, typos, format issues...). My main comments are:

Comment 1. I expect the authors to provide more experimental data on the vitrification of ZIF-62 membranes on different liquid metals, such as SEM/optical images of the prepared glassy ZIF-62 prepared on different liquid metal substrates. Also, I expect a quantitative explanation of the relationship between the surface tension of the liquid metals and the quality of the resulting glassy ZIF-62 membranes.

Response: Following the reviewer's suggestion, we have now added optical images of membranes prepared using Sn, Ga-Sn, and Ga-In substrates. (Figure S10)

Figure S10. Optical (A–C) and laser profilometry (D–F) images of agZIF-62 membranes fabricated on different liquid metal substrates: Sn (A, D), Ga-Sn (10%) alloy (B, E), and Ga-In (25%) alloy (C, F).

To further explore the relationship between the interfacial energy of the liquid metal bath and the vitrification quality of ZIF-62 membranes, we referred to the method presented by Seemann et al. (Physical Review Letter, 2001, DOI: 10.1103/PhysRevLett.86.5534). In that work, the authors quantitatively reconstructed the effective interfacial potential of thin polymer films on solid substrates by analysing the characteristic lateral pattern size (in their case, spinodal wavelength) formed during dewetting. By measuring this lateral length scale from microscopic images and applying the equation:

$$\lambda_s = \left(\frac{8\pi^2\gamma}{-\phi''} \right)^{1/2}$$

where γ is the surface tension of the film and ϕ'' is the second derivative of the interfacial potential, they were able to back-calculate the interfacial energy λ_s that represents film stability.

We found this approach conceptually promising for our own system: molten ZIF-62 films on various liquid metal substrates. Although our membranes do not undergo spinodal dewetting, local thinning and fragmentation during vitrification could in principle reflect to the interfacial energy differences, which might be observable in lateral size across different substrates.

To test this, we attempted with obtaining optical videos of melting ZIF-62 pellets on Ga, Ga-Sn, and Ga-In substrates using confocal scanning laser microscopes (CSLM) integrated IR furnace, and with measuring domain sizes as an analogy to spinodal wavelengths. However, due to the limited image quality at high temperatures (minimum requirement is 723.15 K and high vacuum (10^{-3} Torr) to ensure complete melting and to suppress oxidation), the lateral size could not be obtained with sufficient reliability to reconstruct the interfacial energy. The difficulty is largely due to thermal radiation emitted by liquid metal substrate and therefore overexposure to the camera sensor. We don't have access to more advanced imaging technique and would welcome the reviewer suggests the potential contacts or alternative method for this.

Therefore, despite the physical reasoning from Seemann et al. provides a strong theoretical basis for such analysis, we decided not to include this discussion in the main text due to the lack of quantitative support from our results. We believe this approach would be revisited in future work using more advanced imaging techniques to better resolve vitrification dynamics and interfacial effects.

Comment 2. Is the proposed method applicable for other glass MOFs (e.g., ZIF-4 & TIF-4)? As the surface energy of the melted ZIF-4 or other MOFs may be different from that of ZIF-62, which liquid metal could be the best for the preparation of high-quality glass MOF membranes? Why?

Response: We thank the reviewer for this thoughtful question regarding the broader applicability of the proposed method. In this study, we focused on ZIF-62 due to its well-established glass-forming behaviour, including extrapolated surface tension from experimental measurements and viscosity at molten state. While the concept of using liquid metal baths to facilitate vitrification is in principle generalisable to other meltable MOFs (such as ZIF-4 or TIF-4), we acknowledge that further extension would require careful evaluation of thermodynamic and interfacial factors, such as melt surface tension, viscosity, chemical compatibility, thermal stability, and decomposition risk, because these parameters critically influence spreading, wetting, and resulting membrane morphology.

We position our study as a proof-of-concept for applying the float glass-inspired process to porous meltable material systems more broadly. Future work is needed to identify the optimal liquid metal for each MOF system based on these criteria. Hence, we added the following paragraph for clarification in the revised manuscript, "Beyond ZIF-62, the applicability of this interface-assisted vitrification strategy to other MOF glasses remains an open question. In particular, the selection of liquid metal for each MOF system would require further study of melt-bath interfacial properties, including surface tension, wettability, and chemical compatibility, which are not addressed in this work."

Comment 3. Can the authors provide detailed information on determining the proportion of the zni impurities?

Response: Following the reviewer's suggestion, we have added detailed Rietveld refinement procedure to estimate the phase composition as a separate supplementary section in the revised Supplementary Information as below:

"The relative phase composition of ZIF-62 and ZIF-zni in the P1-6 sample was estimated via Rietveld refinement of PXRD data using the GSAS-II software package. A two-phase model was adopted, employing crystallographic information files (CIFs) for ZIF-62 and ZIF-zni sourced from Nozari et al. (J. Chem. Phys., 2020, DOI: 10.1063/5.0031941) and Chokbunpiam et al. (Microporous Mesoporous Mater., 2013, DOI: 10.1016/j.micromeso.2012.12.047), respectively.

Refinement parameters included background correction, zero shift, peak profile, and independent scale factors for each phase. The relative weight fractions were calculated using the following equation:

$$W_i = \frac{S_i Z_i M_i}{\sum_j S_j Z_j M_j}$$

Where S_i is the refined scale factor taken from the best fitting by GSAS-II program, Z_i is the number of formula units per unit cell, and M_i is the molecular weight of phase i .

Comment 4. I really appreciate the methylation experiments presented in the work. Can the authors provide more detailed information on the methylated glass ZIF-62 (e.g., Raman mapping? Pore size analysis?)?

Response: We thank the reviewer for their valuable suggestion. To assess the homogeneity of the methylation on agZIF-62, we performed Raman spectroscopy and mapping.

Single-point Raman spectra (Figure S13B) show a clear reduction in the C-N stretching vibration at $\sim 1280 \text{ cm}^{-1}$ in the methylated membrane (Group 3) compared to the untreated (Group 1), indicating successful chemical modification. To visualize spatial uniformity, we conducted 20×20 -point Raman mapping on each sample (Figure S13C). We used the intensity ratio of the 1280 cm^{-1} (C-N) peak to the 1600 cm^{-1} peak (aromatic C=C, unaffected by methylation) as an area indicator. The methylated sample shows a consistently higher and more uniform 1280/1600 ratio across the surface, confirming the homogeneous nature of the modification. These results support successful and spatially uniform methylation of agZIF-62.

Figure S13. Raman analysis of untreated and methylated agZIF-62 membranes. (A) Optical image of the mapped agZIF-62 membrane surface (Group 1, scale bar: 200 μm). (B) Representative Raman spectra from untreated (Group 1, blue) and methylated (Group 3, red) membranes. A distinct reduction of the C–N stretching peak at $\sim 1280\text{ cm}^{-1}$ is observed after methylation. (C) Raman intensity ratio mapping of 1280 cm^{-1} (C–N) to 1600 cm^{-1} (aromatic C=C) over a 20×20 grid of selected area.

We also added the following sentence in the revised manuscript:

“In parallel, Raman spectra confirmed a reduction in the C–N stretching vibration ($\sim 1280\text{ cm}^{-1}$) in Group 3 compared to Group 1, consistent with the presence of methylated imidazole species. Mapping of the $1280/1600\text{ cm}^{-1}$ intensity ratio further revealed a uniform distribution of this chemical change across the membrane surface, serving as a spatial indicator of successful and homogeneous post-synthetic modification (Supplementary Figure S13).”

Additional points

Comment 1. Line 124: The authors reported that the proportion of the generated ZIF-zni impurity varies from the order in which different solutions are added during the solvothermal synthesis. It is relatively easy to understand the differences between P1 and P6. However, I am confused on the differences between P1 and P2. As suggested in Table S1, there is a considerable ZIF-zni proportion difference between the samples obtained from P1 and P2. How could it be?

Response: We thank the reviewer for pointing out this important distinction. Although P1 and P2 both involve complete mixing of all precursors, the key distinction lies in the delayed addition of imidazole in P2. This delay likely results in transient local environments with elevated concentrations of imidazole, which can significantly influence the nucleation behaviour of ZIF species.

Previous studies have shown that the order in which precursors are introduced can play a decisive role in phase selection by inducing local inhomogeneities during the early stages of nucleation (Carpenter et al., *Chemical Society Reviews*, 2023, DOI: 10.1039/D3CS00312D). Furthermore, Zn^{2+} ions are known to preferentially coordinate with more basic or less sterically hindered ligands under certain synthesis conditions (Nam et al., *Small Methods*, 2022, DOI: 10.1002/smt.202200772).

Given that imidazole (Im) is both smaller and more basic than benzimidazole (bIm), we believe that Zn²⁺ ions are more likely to coordinate with Im in these transiently enriched regions, thereby promoting the nucleation and growth of ZIF-zni. This interpretation is consistent with the observed increase in ZIF-zni content in P2 relative to P1.

Comment 2. Line 171-172. The authors claimed that the surface area of the glass ZIF-62 membranes by N₂ is 252 m²/g. As glass ZIF-62 generally shows very little adsorption of N₂, I am curious on what kind of test the authors used for the result?

Response: We thank the reviewer for pointing out this inconsistency. The reported value of 252 m²/g was indeed obtained from CO₂ adsorption at 273 K, not N₂. We acknowledge that this was a typographical error in the original text and have corrected the statement accordingly in the revised manuscript. CO₂ is commonly used to probe the microporosity of glassy ZIFs due to its stronger interaction with the framework and smaller kinetic diameter. The reported surface area is consistent with recent literature on high-purity glassy ZIF-62 samples prepared and activated under comparable conditions.

Comment 3. The authors tried different liquid metals, e.g., gallium, pure tin, gallium-tin (10%), and gallium iridium (25%) for the synthesis of glass ZIF-62 membranes. Do these liquid metals affect the quality of the fabricated membranes? Gallium bath was selected for the following experiments. Why? Detailed material characterizations of the membranes are recommended.

Response: We thank the reviewer for the suggestion. We have successfully fabricated glassy ZIF-62 membranes using all tested liquid metals, including gallium, tin, Ga-Sn, and Ga-Ir alloys. Representative optical images of these membranes have been included (Figure S10) to demonstrate successful vitrification.

Among these, we choose to fully characterise membrane obtained from gallium because it is pure element with well-defined surface energy, also it offers the most representative and reproducible membrane quality along with safety of handling and environmental compatibility. Gallium is non-toxic, has a low vapor pressure, and does not produce hazardous fumes under experimental conditions, making it safer to handle and dispose of compared to other liquid metals such as tin or iridium alloys.

A table summarising the surface energies of the tested metals has been added (Table S2) in the revised Supplementary Information.

Table S2. Surface Tension of Liquid Metals and Ga-Sn Alloys at Melting Point and Elevated Temperature (723 K).

Composition	Surface Tension (at Melting Point) mN/m	Surface Tension (723K) mN/m	Note	Ref
Gallium	700–750, 29.8 °C	542-580	Oxide skin surface tension ~360 mN/m	4
Tin	559–569, 231.5 °C	540–550	-	5,6
Gallistan	524-545, 10.5 °C	~548.8	Surface tension stable across temperature	7,8
Ga-Sn (10%)	~620–640, 20.0 °C	Not available in literature	Oxidizes rapidly; oxidation alters surface tension and composition.	9

Comment 4. As stated by the authors, the thickness of the prepared glass ZIF-62 membranes ranges from 2 μm to tens of micrometers, which is controlled by the quantity of the mother precursor, i.e., crystalline ZIF-62 powders. What are the thicknesses of the crystalline ZIF-62 pellets? Does the thickness of the ZIF-62 pellet influence the thickness of aZIF-62 membrane? Also, is the size of the prepared aZIF-62 membrane the same as the mother crystalline ZIF-62 pellet?

Response: We thank the reviewer for this thoughtful question. In previous reports on glass-forming MOFs, the lateral dimensions of monolithic ZIF crystals generally remain unchanged after vitrification, with occasional contraction attributed to melt viscosity, poor wetting on solid supports, and partial pore collapse (Smirnova, *Nat. Mater.*, 2024, DOI: 10.1038/s41563-023-01738-3).

In our work, the crystalline ZIF-62 precursor is typically pressed into pellets with a thickness of approximately 200–800 μm , estimated based on the precursor mass and pressing conditions. Due to the surface energy matching, thickness of the final membrane from liquid metal bath is not solely determined by the pellet geometry. Instead, it is governed by both the amount of precursor material and its spreading behaviour. Notably, the molten precursor exhibits enhanced surface mobility on gallium, leading to an increase in membrane area compared to the starting pellet. This spreading behaviour is beneficial for producing large-area membranes from limited precursor.

Comment 5. The authors observed that the generated ZIF-zni impurity show negligible effects on the H_2 permeance while has an adverse effect on the CO_2 permeance. Do the authors have any assumptions/proof on this result?

Response: We thank the reviewer for the comment. Our interpretation is based on the distinct transport mechanisms governing these two gases. CO_2 permeation in agZIF-62 membranes is largely driven by adsorption-assisted diffusion, facilitated by uncoordinated nitrogen sites introduced during vitrification. ZIF-zni, in contrast, is a denser ZIF phase with significantly lower CO_2 adsorption capacity, as reported in literature (Frentzel-Beyme, et al., *Nature Communications*, 2022, DOI: 10.1038/s41467-022-35372-5). Its presence thus reduces the population of adsorption sites and interrupts the defect-rich network that supports CO_2 transport.

On the other hand, H_2 permeates primarily through size-selective diffusion due to its small kinetic diameter (2.89 \AA) and low molecular mass and is less sensitive to the presence or absence of adsorption sites. Therefore, the introduction of ZIF-zni has minimal impact on H_2 transport. We have now included a brief discussion of this point in the revised manuscript to clarify that the observed drop in CO_2 permeance arises from the combined effects of reduced adsorption capacity and the potential disruption of defect connectivity in the glass matrix.

The following sentence has been added in the revised manuscript:

“This observation suggests that ZIF-zni may reduce CO_2 permeance by limiting adsorption sites and disrupting the defect network, while having minimal impact on H_2 due to its size-selective diffusion.”

Comment 6. Line 276: The authors argued that the methylation of the uncoordinated sites of aZIF-62 membranes leads to considerably decreased CO_2 permeance, while it has very little influence on $\text{H}_2/\text{N}_2/\text{CH}_4/\text{C}_2\text{H}_6$ (Figure 3C). How could the $\text{H}_2/\text{N}_2/\text{CH}_4/\text{C}_2\text{H}_6$ permeance remain after the methylation process, given that this process introduces a decreased surface area of the aZIF-62 membranes as mentioned by the authors (Line 276).

Response: We thank the reviewer for this insightful observation. We agree that the relatively unchanged permeance of H₂, N₂, CH₄, and C₂H₆ after methylation, despite a reduction in BET surface area, is noteworthy. As discussed in the manuscript, the methylation selectively blocks uncoordinated nitrogen sites, which are primarily involved in CO₂ adsorption due to its high quadrupole moment and strong affinity for electron-rich sites. In contrast, H₂, N₂, CH₄, and C₂H₆ are non-polar or weakly polar molecules that exhibit minimal specific interactions with these sites. Their transport through the membrane is largely governed by size exclusion and diffusivity through the continuous pore network.

Comment 7. It seems the C₂H₄ permeance varies from the Group 1-3 samples. The authors claimed that the methylation has negligible effects on the crystalline ZIF-62 powders since they have few uncoordinated sites. However, why is there a C₂H₄ permeance change in the Group 2 samples compared to the Group 1 samples?

Response: We thank the reviewer for drawing attention to the C₂H₄ permeance differences. As shown in Figure 3C, there is a slight variation in the C₂H₄ permeance between Group 1 and Group 2. We note that this variation falls within the experimental uncertainty range of our measurement setup and may also reflect subtle differences in precursor packing or spreading during vitrification. While Group 2 was treated with (TMS)CHN₂ prior to vitrification, the methylation was not successful—as confirmed by the absence of 1-methylimidazole signals in NMR. These characterisations indicates that the overall coordination structure of Group 2 remained similar to Group 1. Therefore, we do not attribute this difference to chemical modification.

This explanation is supported by prior studies using solid-state NMR on ZIF materials, which showed that ethylene (C₂H₄) is more sensitive than ethane (C₂H₆) to subtle changes in pore geometry, resulting in observable selectivity shifts even when bulk framework composition remains similar (Xiao, Y., Chemistry A European Journal, 2021, DOI: 10.1002/chem.202101779).

Some minor issues are:

Comment 1. A table summarizing the surface energies of the used liquid metals is recommended.

Response: A table has been added to the SI Table S2.

Table S2. Surface Tension of Liquid Metals and Ga-Sn Alloys at Melting Point and Elevated Temperature (723 K).

Composition	Surface Tension (at Melting Point) mN/m	Surface Tension (723K) mN/m	Note	Ref
Gallium	700–750, 29.8 °C	542–580	Oxide skin surface tension ~360 mN/m	4
Tin	559–569, 231.5 °C	540–550	-	5,6
Gallistan	524–545, 10.5 °C	~548.8	Surface tension stable across temperature	7,8
Ga-Sn (10%)	~620–640, 20.0 °C	Not available in literature	Oxidizes rapidly; oxidation alters surface tension and composition.	9

Comment 2. The unit of the permeance in Table S2 is missing.

Response: The unit of permeance has been updated to be GPU. The table number has been updated to S3 in the revised SI.

Comment 3. The lettering used for figure numbers in the article is sometimes uppercase and sometimes lowercase. Please make it consistent. Line 165. Figure 2f.

Response: The consistency has been confirmed.

Comment 4. The font of the figure legends should be modified.

Response: The font in Figure 1F legend has been corrected for consistency.

Comment 5. The details in determining the degree of crystallinity of the synthesized ZIF-62 powders should be given.

Response: We have added detailed Rietveld refinement procedure to estimate the phase composition in the supplementary section of the revised version.

Comment 6. Line 191: Where is Supplementary Note 1?

Response: It has been corrected to “Simulation Note in Supplementary Text”

Comment 7. Line 292: The authors mentioned “This structural motif is consistent with PALS results (Figure 2F) ” The attached Figure 2F doesn’t show any data on PALS results. Please correct this. Is this Figure 1F?

Response: The error has been corrected, Figure 1F bears the relevant PALS results.

8. Line 275, where’s Figure 3X? This figure is missing.

Response: The error has been corrected.

Reviewer #3 (Remarks to the Author):

Response: We thank the reviewer for their contribution and support of the initiative to engage and recognise early career researchers in the peer review process. We appreciate the time and effort dedicated to this manuscript.